# The Effect of Creatine Nitrate and Caffeine Individually or Combined on Exercise Performance and Cognitive Function: A Randomized, Crossover, Double-Blind, Placebo-Controlled Trial

**DOI:** 10.3390/nu16060766

**Published:** 2024-03-07

**Authors:** Gina Mabrey, Majid S. Koozehchian, Andrew T. Newton, Alireza Naderi, Scott C. Forbes, Monoem Haddad

**Affiliations:** 1Department of Kinesiology, Jacksonville State University, Jacksonville, AL 36265, USA; gmabrey@jsu.edu (G.M.); atnewton@jsu.edu (A.T.N.); 2Department of Sport Physiology, Islamic Azad University, Boroujerd 1706294, Iran; naderi.alireza1366@gmail.com; 3Department of Physical Education Studies, Faculty of Education, Brandon University, Brandon, MB R7A 6A9, Canada; forbess@brandonu.ca; 4Physical Education Department, College of Education, Qatar University, Doha P.O. Box 2713, Qatar; mhaddad@qu.edu.qa

**Keywords:** caffeine, creatine nitrate, ergogenic aids, exercise performance, cognitive function, resistance training, sports nutrition, dietary supplements

## Abstract

This study examined the effect of creatine nitrate and caffeine alone and combined on exercise performance and cognitive function in resistance-trained athletes. In a double-blind, randomized crossover trial, twelve resistance-trained male athletes were supplemented with 7 days of creatine nitrate (5 g/day), caffeine (400 mg/day), and a combination of creatine nitrate and caffeine. The study involved twelve resistance-trained male athletes who initially provided a blood sample for comprehensive safety analysis, including tests for key enzymes and a lipid profile, and then performed standardized resistance exercises—bench and leg press at 70% 1RM—and a Wingate anaerobic power test. Cognitive function and cardiovascular responses were also examined forty-five minutes after supplementation. Creatine nitrate and caffeine that were co-ingested significantly enhanced cognitive function, as indicated by improved scores in the Stroop Word–Color Interference test (*p* = 0.04; effect size = 0.163). Co-ingestion was more effective than caffeine alone in enhancing cognitive performance. In contrast, no significant enhancements in exercise performance were observed. The co-ingestion of creatine nitrate and caffeine improved cognitive function, particularly in cognitive interference tasks, without altering short-term exercise performance. Furthermore, no adverse events were reported. Overall, the co-ingestion of creatine nitrate and caffeine appears to enhance cognition without any reported side effects for up to seven days.

## 1. Introduction

Athletes often use dietary supplements to enhance their training and exercise performance. In recent years, sports nutrition has become a more nuanced science concerning performance enhancement, issues of safety, and legality [1]. Among the extensively studied dietary supplements, caffeine and creatine independently are well supported to have ergogenic properties [2,3]; however, the potential synergistic effects from their combined use are limited [4,5]. In sports, caffeine has been well documented to improve maximum strength and muscular endurance [6] as well as anaerobic performance [7], revealing its effects on the central nervous system [2]. Caffeine’s resemblance to adenosine in molecular structure enables it to attach to adenosine receptors, particularly targeting the A2A subtype. This attachment diminishes sensations of pain and weariness while concurrently enhancing neural excitability [8]. Caffeine consumption can lead to physiological effects such as increased heart rate (HR), blood pressure (BP) [9], bronchodilation [10], and reduced perceived exertion during strenuous activities [2]. Caffeine’s ability to enhance cognitive function, such as attention, vigilance, and reaction time, has made it a valuable supplement for athletes in sports that require sustained concentration and quick decision making [11].

The effect of creatine on athletic performance has been a focal point of sports nutrition research since the 1990s [12]. Creatine’s role in rapidly replenishing adenosine triphosphate (ATP) during anaerobic exercise is well established, with numerous studies corroborating its efficacy in improving short-term power output and facilitating greater training volume [13]. This translates to improvements in strength, sprinting ability, and muscle mass [14]. Beyond its physical benefits, creatine’s influences on brain energy metabolism have led to new avenues of research into its potential cognitive benefits, particularly under high energy demand or stress conditions [15,16].

The synthesis of creatine nitrate (i.e., C4H10N4O5) introduces a compound that promises the benefits of creatine monohydrate and posits the added advantage of enhanced nitric oxide production [17]. This particular molecule is of interest to athletes, as nitric oxide is pivotal in modulating blood flow, muscle oxygenation, and nutrient delivery. While studies have begun to explore the ergogenic potential of creatine nitrate, much remains to be understood about its efficacy and the mechanisms through which it may act [18]. Creatine nitrate is typically produced via a chemical reaction that involves mixing creatine with nitric acid in the presence of water. This leads to the crystallization of either creatine dinitrate or creatine trinitrate. An alternative method involves the use of nitrous acid in place of nitric acid, resulting in the formation of creatine nitrate [19]. Earlier studies reported that creatine nitrate supplementation increased circulation and muscle creatine levels from 5 to 28 days [20,21] and improved body composition and strength performance as much as creatine monohydrate during 28 days [21]. Moreover, Dalton et al. 2017 reported that 3 and 6 g creatine nitrate over five days improved one-repetition maximum (1RM) and muscular endurance bench press performance at 70% of 1RM without an ergogenic effect during a 4 km time trial performance [22]. Creatine nitrate also seems safe when ingested at 1 to 6 g/day doses for up to 28 days [20,21,22].

The interactions between caffeine and creatine are mixed and remain to be fully elucidated. Some studies have assessed the potential synergistic effects that may emerge from their combined use when caffeine is acutely supplemented after a creatine-loading phase [4,5]. At the same time, some have proposed that caffeine may hinder creatine’s action due to its opposite effect on muscle relaxation time [23] and potential gastrointestinal distress [24,25]. Others suggest that these effects may not significantly impact the overall ergogenic potential of creatine [26]. For instance, Pakulak et al. (2022) did not report significant effects in variables related to body composition and resistance exercise performance followed by six weeks of resistance training co-ingestion of creatine monohydrate with caffeine supplementation compared to their isolated ingestion and placebo [27]. Considerations for concurrent supplementation indicate that while creatine and caffeine are popular ergogenic aids and have been shown to improve aspects of high-intensity exercise performance, there is still debate and inconclusive evidence regarding their combined effects during resistance exercise [27,28].

The study’s primary aim was to evaluate the effects of a seven-day high-dose intake of (1) caffeine, (2) creatine nitrate, and (3) their combination on intense intermittent exercise performance and mental focus in resistance-trained males. Secondary outcomes included assessing the impact of these supplements on cardiovascular hemodynamics and blood metabolite levels. This study sought to understand how these substances, alone or in combination, can enhance physical performance and cognitive function and their physiological effects during intense exercise. We hypothesized that ingesting caffeine creatine nitrate alone would improve mental focus, resistance exercise performance, and related performance indices (e.g., peak power, mean power, total work, etc.); however, more improvements would occur because of supplementation with the combined caffeine and creatine nitrate. We further hypothesized that ingesting either individual or combined creatine nitrate and caffeine supplements would not adversely affect the hepatorenal function or hemodynamic indices following acute and short-term ingestion.

## 2. Materials and Methods

### 2.1. Study Overview and Ethical Considerations

This study was carried out at the Human Performance Laboratory (HPL) of Jacksonville State University. It received the requisite approval from the university’s ethics committee, ensuring compliance with all ethical standards for research. The study’s design and progression are depicted in Figure 1, which provides a Consolidated Standards of Reporting Trials (CONSORT) flow diagram, and Figure 2 details the methodological rigor.

### 2.2. Study Protocol

#### 2.2.1. Recruitment and Familiarization

Participants were recruited via email campaigns and campus-wide flyers. Interested individuals were required to attend a familiarization session, where they completed informed consent forms, provided a comprehensive health history reviewed by a registered nurse, and received verbal and written details regarding the study protocol. Additionally, participants practiced the exercise tests to be used in the study. Eligibility was determined based on standard anthropometric measurements, including height, weight, HR, BP, and body fat mass. This was followed by a 1RM strength test for bench and leg press exercises, adhering to the National Strength and Conditioning Association’s guidelines [29]. A Wingate 30 s anaerobic capacity test using a cycle ergometer was also conducted. To minimize potential confounders, all assessments were completed in a single session, simulating the conditions of a typical workout. Successful completion of this session led to subsequent baseline evaluations and randomization of the study.

#### 2.2.2. Eligibility Criteria

Participants were eligible for the study if they had at least two years of experience in multi-joint resistance training. We excluded individuals under 18 or over 40, those with metabolic disorders such as diabetes, cardiovascular diseases, arrhythmia, thyroid conditions, and those on prescription medications. Additionally, individuals with a BMI outside the range of 18.5 to 24.9, indicating underweight or obesity, were also excluded. Ineligibility criteria also included known intolerance to caffeine or natural stimulants, smokers, and those consuming over 12 alcoholic beverages weekly. Participants were instructed to maintain their current training and dietary habits without introducing new exercises or nutritional changes for the duration of the study.

#### 2.2.3. Performance Testing Protocol

Participants were subjected to baseline and subsequent evaluations under controlled conditions, which included a 12 h fasting period and a 48 h abstention from exercise, caffeine, and certain medications or supplements. A 20 mL venous blood sample was collected upon arrival at the laboratory. Participants ingested their assigned supplements following anthropometric measurements and baseline recordings of HR and BP. Following a 45 min pause following supplementation, participants completed the Stroop Word–Color test and exercise performance assessments. The suggestion to consume caffeine about 45 min before exercise is supported by research and practice as being sufficient to demonstrate improvement in various aspects of exercise performance, such as strength, power, endurance, and recovery [6,30]. The ideal time for creatine ingestion in relation to exercise to optimize muscle loading and performance gains is not well established [31].

After completing the cognitive function test, the Visual Analog Scale (VAS) assessed participants’ readiness to perform. In addition, they were asked to complete comprehensive questionnaires to evaluate their sleep quality, caffeine consumption, and any side effects they may have experienced.

The physical testing commenced with a standardized warm-up, leading to targeted resistance exercises—bench and leg presses—performed for three sets at 70% 1RM, with the final set pushing for maximal repetitions [32]. HR and BP were monitored closely post-exercise. A subsequent five-minute rest prefaced the Wingate anaerobic test, which was executed to quantify anaerobic performance indicators such as peak power, mean power, and fatigue index [33]. Cardiovascular responses were again evaluated following this test.

#### 2.2.4. Supplementation

For the intervention, four distinct treatments were administered in this study: creatine nitrate [CN: (4 g creatine; 1 g nitrate), 5 g/d + 0.675 g/d maltodextrin)], caffeine (CAF: 400 mg/d + 5 g/d maltodextrin), a combination of both CN and CAF (CO: 5 g/d creatine nitrate + 400 mg/d caffeine), and a placebo (PL: 5.4 g/d maltodextrin). Each treatment was administered 45 min prior to the experimental testing. During the week of supplementation, aside from baseline and follow-up sessions, participants consumed their specified supplement in two different schedules: about 45 min pre-workout on exercise days and around noon on days without exercise. Following a week-long washout period, they switched supplements and repeated the process in a randomized fashion. Bulk Supplements (Henderson, NV, USA) produced the CAF and maltodextrin, and the CN was sourced from APS Distribution, Inc. (Norcross, GA, USA). In this study, the dosages of creatine nitrate, caffeine, and their combination were based on previous research [22,30,34], which has shown these specific amounts to enhance exercise performance and cognitive function in athletic populations effectively. This rationale underpins our selection of dosages to investigate their impact on similar outcomes. To ensure accuracy and consistency in dosing, each supplement was precisely measured. The study maintained its double-blind design using identically encapsulated supplements distributed in coded containers. Supplement intake was aligned with the participants’ exercise schedules, incorporating a 7-day washout period before transitioning to a different treatment in a randomized sequence [6,33].

#### 2.2.5. Anthropometry

Initial anthropometric measurements were conducted precisely: height and weight were measured to the nearest 0.01 m and 0.1 kg using the Health-O-Meter 349KLX Medical Weight Scale.

#### 2.2.6. Dietary Monitoring

Throughout the study, participants meticulously documented their dietary intake. Comprehensive records included all foods and beverages consumed, focusing on brand names, preparation methods, and exact quantities. For accurate dietary logging, participants used the MyFitnessPal app, completing four sets of three-day food diaries. A certified dietitian provided initial training on portion size estimation and comprehensive instructions for effectively using the dietary tracking application.

#### 2.2.7. Stroop Word–Color Test

The Stroop Word–Color Test was utilized to assess the effects of dietary supplementation on attentional control, processing speed, and cognitive flexibility [35,36,37,38,39]. The assessment consisted of three tasks: identifying the color of words printed in black, naming the color of Xs, and stating the color of words printed in incongruent colors. Participants were instructed to complete these tasks as quickly and accurately as possible, with performance gauged by the number of correct responses within a specified time.

#### 2.2.8. Readiness to Perform Exercise

To ascertain participants’ readiness for physical activity, we employed the VAS. This subjective measure required participants to indicate their level of agreement with six statements concerning their current physical and mental state, ranging from sleep quality to muscle soreness, on a 20 cm line that represented a continuum from “strongly disagree” to “strongly agree”.

#### 2.2.9. Muscular Strength and Wingate Anaerobic Power

Maximal strength was measured after a structured warm-up. This began with ten reps at 50% of the predicted 1RM, followed by five reps at 70%, and one rep at 90%. Participants gradually increased the weight until their accurate 1RMs were identified. They were urged to ascertain their 1RM during the familiarization session. Once established, participants underwent three sets of bench and leg press evaluations. In both the first and second sets, they lifted 70% of their 1RM for 10 reps on both exercises, with 2 min rest intervals between sets and a 5 min break between different testing exercises. In the third set, they completed as many reps as possible. The total lifting volume was determined by multiplying the weight by the number of successful reps. Throughout the tests, verbal encouragement was given to bolster maximum effort. All strength evaluations were conducted following established protocols and using the bench press and hip/leg sled machines (Hammer Strength, Schiller Park, IL, USA) [40]. The Wingate test involved participants pedaling on a specific bike ergometer as hard as possible for 30 s. The Wingate anaerobic power test was conducted using a calibrated, friction-loaded cycle ergometer (Monark 894E Peak bike, MONARK, Vansbro, Sweden). Each participant’s work rate was standardized at 7.5 J/kg/rev. Before initiating the main test, participants were instructed to achieve a speed of 70 RPM and maintain it over five brief intervals, each lasting five seconds. After this warm-up, they were prompted to sprint at their utmost capacity for 30 s. Throughout this sprinting phase, verbal support was provided as an external motivator to elicit the participant’s highest possible performance [33].

#### 2.2.10. Blood Chemistry Assessment

Before the baseline and follow-up testing sessions, participants abstained from exercise, CAF, and non-prescription stimulants for 48 h. Additionally, they fasted for 12 h prior to providing a blood sample. Approximately 20 mL of blood was drawn from the antecubital vein in the forearm, adhering to standard blood collection methods. This blood was collected into two 7.5 mL BD Vacutainer serum separation tubes (Becton, Dickinson, and Company, Franklin Lakes, NJ, USA). Post-collection, the EDTA-treated tubes were carefully inverted ten times, allowed to sit for 15 min at room temperature, and centrifuged by Drucker 642E Labcorp Horizon Mini E Centrifuge (Philipsburg, PA, USA). This was carried out at 3500 rpm for 10 min to segregate plasma and blood cells. The isolated serum samples were then sent to the LabCorp facility at 4 °C for further examination [41].

The blood samples were assessed for creatine kinase (CK), lactate dehydrogenase (LDH), alkaline phosphatase (ALP), aspartate aminotransferase (AST), and alanine aminotransferase (ALT) using the Roche Hitachi Modular analyzer (Roche Diagnostics, Indianapolis, IN, USA). A comprehensive lipid profile was evaluated to measure total cholesterol (TC), high-density lipoproteins (HDL), low-density lipoproteins (LDL), very-low-density lipoproteins (VLDL), and triglycerides (TG). TC and TG levels were quantified using enzymatic colorimetric assays, whereas HDL levels were ascertained following precipitation with sulfated α-cyclodextrin in an alkaline magnesium chloride medium. LDL and VLDL concentrations were estimated utilizing the Friedewald equation, which is in agreement with established methodologies [42].

Furthermore, another blood sample was stored in a 3.5 mL BD Vacutainer tube with a lavender cap containing K2 EDTA. This was kept at room temperature for a short while and later refrigerated for a few hours, after which a comprehensive blood count was performed. The analysis encompassed parameters such as hemoglobin, hematocrit, red blood cell (RBC) counts, and others and was executed using the Abbott Cell Dyn 1800 (Abbott Laboratories, Abbott Park, IL, USA) automated hematology analyzer.

#### 2.2.11. Sleep Quality, Caffeine Tolerance Assessment, and Side Effect Questionnaires

At each evaluation, participants completed a brief questionnaire designed to appraise sleep quality and discern any disturbances potentially linked to the supplementation. This instrument gauged various facets of sleep, such as duration, perceived quality, vitality upon waking, and any disruptions experienced within a 48 h window, thereby distinguishing between suboptimal and satisfactory sleep quality. Additionally, a caffeine tolerance survey was administered, prompting individuals to quantify any experienced symptoms—such as lethargy, fatigue, or irritability—using a scale ranging from 0 (none) to 4 (severe). Concurrently, participants were asked to complete a side effects questionnaire to identify adverse reactions to the supplements. This tool aimed to record any unwanted symptoms or reactions attributed to the supplementation regimen. Participants evaluated the presence and intensity of potential side effects, such as gastrointestinal discomfort, headaches, or palpitations, using a similar scale that ranged from 0 (absent) to 4 (severe). This systematic approach allowed for a comprehensive assessment of the supplement’s tolerability and safety profile alongside evaluating sleep quality and caffeine tolerance.

### 2.3. Statistical Analysis

A 2-factor (time × treatment) repeated measures ANOVA was utilized to discern the effects of different treatments over various time points. This approach allowed for a nuanced detection of the treatments’ impact and their temporal interactions. We calculated effect sizes using the partial eta-squared statistic, categorizing effect sizes as small (0.01 ≤ partial eta-squared < 0.06), medium (0.06 ≤ partial eta-squared < 0.14), and large (partial eta-squared ≥ 0.14), providing insight into the magnitude of differences between conditions. When significant effects were identified, Tukey’s post hoc tests elucidated specific contrasts between treatments. Additionally, mean changes from baseline and their 95% confidence intervals were reported, offering a precise estimation of population mean differences. Side effects categorized data were analyzed using chi-squared tests, with statistical significance denoted by *p* < 0.05. Results were presented as mean ± standard deviation, reflecting the data’s central tendency and dispersion, thereby upholding analytical rigor and interpretative validity. All statistical analyses were conducted with SPSS Statistics V29 (SPSS Inc., Chicago, IL, USA). GraphPad Prism Version 10.1.1 (GraphPad Software, San Diego, CA, USA) was employed for graph generation.

## 3. Results

### 3.1. Participants

Fifteen resistance-trained males expressed interest in participating. After informed consent was obtained and eligibility assessed, three individuals were excluded—one due to medical screening results and two due to scheduling conflicts. This left 12 participants who completed all four conditions of the study. The demographic and baseline nutritional profiles of these individuals are summarized in Table 1. Table 2 presents the data from a three-day dietary analysis across different treatments. MANOVA indicated no significant differences between treatments in terms of relative energy intake (*p* = 0.07), protein intake (*p* = 0.08), carbohydrate intake (*p* = 0.43), and fat intake (*p* = 0.09). Hydration and dietary practices were monitored and remained stable throughout the study, and the sequence of treatment administration showed no significant impact on the measured outcomes.

### 3.2. Primary Outcomes

#### 3.2.1. Exercise Performance

In Table 3, we outlined our exercise performance outcomes. No statistically significant interactions were identified across any of the performance parameters.

##### Bench Press

Neither repetitions nor lifting volume displayed significant treatment interactions for the bench press. These measures’ effect sizes were 0.145 and 0.141, respectively, indicating a large effect. Mean changes from baseline with 95% CIs were significant for repetitions to failure in CAF (2.16 counts, [0.98–3.34]) and CO (2.16 counts, [0.98–3.34]) versus PL (0.41 counts, [−0.76 to 1.59]) (*p* = 0.04) (Figure 3). The change in bench press lifting volume from baseline to follow-up was PL (22.9 kg, [−71.3 to 117.2]), CN (109.1 kg, [14.8 to 203.4]), CAF (152.1 kg, [57.7 to 246.3]), and CO (155.7 kg, [61.4 to 250.0]).

##### Leg Press

For the leg press, neither repetitions nor lifting volume showed significant treatment interactions. The effect sizes for these measures were 0.092 and 0.090, respectively, both denoting a medium effect. The alterations in leg press repetitions to failure from baseline to follow-up were as follows: PL (1.25 counts, [−3.51 to 6.0]), CN (4.5 counts, [−0.25 to 9.25]), CAF (−1 counts, [−5.75 to 3.75]), and CO (4.66 counts, [−0.08 to 9.42]). Similarly, the changes in leg press lifting volume from baseline to follow-up were PL (−219.3 kg, [−1688.7 to 1250.1]), CN (1144.8 kg, [−324.5 to 2614.3]), CAF (−299.1 kg, [−1768.6 to 1170.2]), and CO (1117.5 kg, [−351.8 to 2587.0]).

##### Wingate Testing

Within the treatment groups, significant alterations were consistently absent across key performance indicators, including peak power, mean power, minimum power, total work, and fatigue index. The data revealed stability in these metrics throughout the treatment’s duration. An effect size of 0.082 indicated a slight effect for peak power, suggesting only minimal differences between treatments in terms of maximum power generation. In the case of mean power, an effect size of 0.258 was noted, slightly higher than that for peak power. Nevertheless, this underscores that, while there may be some variation in average power output across conditions, these differences are insignificant. The effect size for minimum power stood at a notably low 0.043, accentuating the minute variation between conditions when evaluating the smallest power outputs. In relation to total work, an effect size of 0.228 was observed, hinting at slight differences in work executed across conditions, but these differences were not pronounced. Lastly, the fatigue index presented an effect size of 0.063, suggesting a uniform fatigue manifestation rate across different treatments, thus underlining the negligible variation in fatigue onset.

Specifically, the change in peak power for the Wingate test from baseline to follow-up was as follows: PL (3.67 W, [−61.3 to 68.6]), CN (14.6 W, [−50.3 to 79.5]), CAF (−35.1 W, [−100.1 to 29.8]), and CO (27.6 W, [−37.3 to 92.6]). The change in mean power for the Wingate test from baseline to follow-up was as follows: PL (6.04 W, [−14.4 to 26.5]), CN (16.8 W, [−3.68 to 37.3]), CAF (−25.1 W, [−45.6 to −4.63]), and CO (13.5 W, [−6.99 to 34.0]). The change in minimum power for the Wingate test from baseline to follow-up was as follows: PL (7.47 W, [−30.7 to 45.6]), CN (9.97 W, [−28.2 to 48.1]), CAF (3.45 W, [−34.7 to 41.6]), and CO (−16.5 W, [−54.7 to 21.6]). The change in total work for the Wingate test from baseline to follow-up was as follows: PL (348.6 J, [−229.1 to 926.4]), CN (443.6 J, [−134.1 to 1021.4]), CAF (−698.1 J, [−1275.9 to −120.3]), and CO (261.8 J, [−315.9 to 839.6]). Lastly, the change in the fatigue index from baseline to follow-up was as follows: PL (−1.95%, [−7.09 to 3.19]), CN (−0.49%, [−5.63 to 4.65]), CAF (−1.91%, [−7.05 to 3.23]), and CO (2.01%, [−3.12 to 7.15]).

#### 3.2.2. Stroop Testing and Readiness to Perform

The Stroop–Color interference test analysis yielded statistically significant results (*p* = 0.04), highlighting a notable interaction effect. Further examination via post hoc analysis revealed a meaningful distinction in performance between the CO treatment and CN, with the CO treatment demonstrating a superior mean score (CO: 65.9 ± 17.3) compared to the CN treatment (CN: 56.8 ± 10.8) (*p* = 0.007). This difference represents a medium-to-large effect size of 0.163, suggesting that the combined supplementation has a robust impact on cognitive processing as assessed by this specific test. In the Stroop Word and Color tests, the data did not reveal any significant interaction effects. The Word and Color test effect size was quantified at 0.014 and 0.058, respectively. These values denote that the influence of the treatments on these individual components of the Stroop test ranged from small to moderate. Such effect sizes suggest a marginal, although not statistically significant, impact of the supplementation on the cognitive tasks of Word recognition and Color identification when evaluated independently. Moreover, no significant treatment interactions were detected in the total counts of the Stroop test, which is further substantiated by a small effect size of 0.027. The absence of notable differences among other conditions underscores the specificity of the cognitive impact on the interference aspect of the task, which is integral to understanding the nuanced cognitive enhancements provided by the supplementation protocols studied (Table 4). Figure 4 illustrates mean changes from baseline, along with 95% CIs, for the significant changes observed from baseline to follow-up in Word, Color, Word–Color, or summated counts. Significant changes in the Word–Color test were noted between CO (7.41 counts, [2.41 to 12.4]) and CAF (0.25 counts, [−4.78 to 5.24]) treatments (*p* = 0.04). The Stroop Word test exhibited changes from baseline to follow-up as follows: PL (1.66 counts, [−3.83 to 7.16]), CN (3.25 counts, [−2.24 to 8.74]), CAF (3.41 counts, [−2.08 to 8.91]), and CO (1.66 counts, [−3.83 to 7.16]). For the Stroop Color test, changes from baseline to follow-up were observed as follows: PL (2.75 counts, [−0.78 to 6.28]), CN (3.58 counts, [0.05 to 7.11]), CAF (0.5 counts, [−3.03 to 4.03]), and CO (0.75 counts, [−2.78 to 4.28]). Regarding the Stroop total counts, changes from baseline to follow-up were noted as follows: PL (12.9 counts, [2.41 to 23.4]), CN (9.66 counts, [−0.84 to 20.1]), CAF (4.16 counts, [−6.34 to 14.6]), and CO (9.83 counts, [−0.67 to 20.3]). No significant interactions or main effects were observed for the readiness to perform the questionnaire or sleep quality (*p* > 0.05).

### 3.3. Secondary Outcomes

#### 3.3.1. Cardiovascular Hemodynamics

The application of multivariate analysis yielded Wilks’ Lambda values, which indicated the absence of significant differences across the factors evaluated. Specifically, there were no significant statistical changes over time (*p* = 0.61), across treatments (*p* = 0.25), or within the time × treatment interaction (*p* = 0.71). The metrics, including HR and BP, remained consistent pre- and post-exercise, irrespective of the treatment administered (*p* > 0.05).

#### 3.3.2. Blood Chemistry

Throughout the study, blood chemistry analysis from baseline to follow-up demonstrated stability within each treatment group and revealed no significant differences between treatments, as detailed in Table 5, Table 6 and Table 7. Blood parameters remained within standard clinical ranges, and any deviations observed at follow-up were evenly distributed across all treatment groups, including the PL, indicating no significant treatment-related effects on blood chemistry.

#### 3.3.3. Safety Profiles

In our monitoring approach, we utilized in-depth questionnaires to track a wide spectrum of symptoms, encompassing both physical and psychological aspects:

##### Caffeine Questionnaire

Our research incorporated the established Caffeine Questionnaire, a validated instrument employed in numerous preceding studies [33,43,44]. This questionnaire is adept at delineating both the occurrence and magnitude of specific caffeine-induced physiological effects, including but not limited to jitteriness, cardiac palpitations, and gastrointestinal disturbances. Leveraging this tool, our study conducted a profound analysis, revealing intricate correlations between individual caffeine consumption patterns and the diverse spectrum of adverse symptoms reported, thus contributing to a more comprehensive understanding of caffeine’s complex physiological effect within our study cohort.

##### Assessment of Sleep Quality

A short sleep quality questionnaire was used to examine the effect of caffeine on sleep quality during each testing session. The questionnaire is a comprehensive tool to assess sleep habits over the last 48 h. It gathers detailed information about bedtime routines, time taken to fall asleep, wake-up times, and actual sleep duration. Additionally, it probes into sleep disturbances, overall sleep quality, daytime enthusiasm, and the potential impact of a bed partner or roommate on sleep patterns. This approach comprehensively evaluates the various factors contributing to sleep quality and efficiency.

##### Side Effects Questionnaire

This questionnaire is used to monitor and quantify any adverse reactions participants may have experienced due to supplement use. This comprehensive instrument allows participants to rate the frequency and severity of various symptoms such as dizziness, headaches, heart palpitations, shortness of breath, nervousness, and blurred vision. The questionnaire’s design facilitated a systematic collection of data crucial for evaluating the safety profile of the supplements under investigation.

## 4. Discussion

Our research was rigorously designed to examine the effects of CAF and CN alone and in combination on cognitive function and exercise performance among resistance-trained athletes. This study examined numerous variables, including anaerobic exercise performance, attention interference and cognitive processing, readiness for performance, cardiovascular function, and blood chemistry. While individual studies have reported distinct ergogenic benefits of both creatine [45] and caffeine [7], the combined effects of these compounds are limited.

### 4.1. Primary Outcomes: Exercise Performance and Cognitive Function

Our findings are in agreement with Jagim et al., 2016, who reported variable responses to the combined supplementation [46]. Given that significant differences from the placebo were found only for BP reps to failure at 70% 1RM in the CAF and CO conditions and total work and mean power for the CAF group, all supplement types failed to demonstrate clear performance enhancement across all or most outcomes. It is possible that this suggests a complex interplay between caffeine and creatine [33]. Trexler et al. (2015) indicated that concurrent CAF supplementation may diminish the effectiveness of creatine, even though some performance benefits are noted with their combined use [28]. This underscores the subtle complexities of how supplements interact and the consequential effects on exercise performance. Our study contributes to the growing body of evidence that highlights a personalized approach to pre-workout supplements (PWS), considering the multifaceted nature of their effects on physical exertion and cognitive function.

The observed improvements from baseline in the Stroop Color and Word–Color interference tests following CO supplementation align with existing research demonstrating the cognitive benefits of creatine. Previous investigations have indicated that creatine supplementation can elevate brain creatine levels, potentially enhancing cognitive performance, particularly in tasks requiring memory and executive function [47,48]. The systematic review of randomized controlled trials demonstrates creatine’s positive impact on healthy individuals’ memory performance, which supports the observed outcomes in our study [47]. The Stroop tasks, often used to measure cognitive processes such as attention, processing speed, and executive function, are susceptible to cognitive performance changes. The improvements in these tasks within our study could indicate creatines’ role in augmenting cognitive processing capabilities. Indeed, studies have demonstrated that creatine supplementation is associated with improved neuropsychological performance, particularly under conditions of mental stress, such as sleep deprivation or aging, akin to the demands placed on cognitive function during competitive sports [48,49]. Furthermore, the enhanced cognitive performance in the CO group could suggest a synergistic effect of creatine when combined with CAF. This hypothesis is supported by research indicating that the co-supplementation of creatine with CAF may amplify cognitive benefits, with changes potentially related to increased activation in the prefrontal cortex, an area of the brain critical for executive functions [50]. Such combined effects could be particularly advantageous for athletes whose sports require physical prowess, strategic thinking, and rapid decision-making under pressure.

While our results demonstrated enhancements in cognitive function, we did not observe any changes in perceived readiness to perform, as measured by a VAS. Similar to our results, Lutsch et al. (2019) reported no significant enhancements in subjective energy levels after PWS consumption [51]. This suggests that the subjective sense of readiness or fatigue may not always align with cognitive performance improvements. In contrast, the findings of Kedia et al. (2014) demonstrate that consistent PWS over six weeks can lead to marked subjective improvements in energy and concentration, objective increases in focus, and reduced fatigue during physical tests [52]. These findings highlight that the impact of PWS can vary widely depending on the duration of intake, the specific cognitive or physical domains being measured, and individuals’ subjective perceptions.

Our study examined the effects of CAF, CN, and CO on physical performance metrics, including bench press and leg press repetitions and volumes, as well as various Wingate test parameters. Our findings support Wax et al. (2021), who reported variable outcomes with creatine supplementation across different exercise modalities, particularly endurance activities, and scenarios where added body mass could hinder performance [53]. Complementing this, our observations on caffeine’s role in resistance training resonate with the findings of Grgic et al. (2018), which highlight the complexity of caffeine’s ergogenic effects, contingent upon factors like exercise type, dosage, and individual differences. Intriguingly, genetic components such as the CYP1A2 gene variant may further inform these individual differences, offering an additional layer of complexity in the ergogenic response to caffeine supplementation and its impact on exercise performance [8,54]. Furthermore, the Wingate test results indicated consistent performance across indicators such as peak power, mean power, minimum power, total work, and fatigue index, echoing the findings of Mielgo-Ayuso et al. (2019) that suggest caffeine’s effects on anaerobic performance are not as pronounced as once thought [55]. Despite moderate ESs, this stability challenges the notion of a straightforward ergogenic effect of CN and CAF. It is consistent with the nuanced perspectives Guest et al. (2021) presented on the complexity of interpreting performance outcomes from such supplements [30]. The absence of significant treatment interactions in our study, despite observable changes in mean values, contributes to the ongoing discussion on the role of these supplements in exercise performance enhancement. This underscores the importance of nuanced interpretation of ergogenic aids, as Hall et al. (2021) posited, suggesting that these supplements may have more subtle and context-dependent effects on physical performance than previously understood [56]. Our comprehensive analysis endorses a cautious approach to declaring the efficacy of CN and CAF as performance enhancers. It supports the call for further investigations, such as those recommended by Kreider et al. (2017), to unravel the intricate interactions influencing exercise outcomes [3].

Future investigations should focus on the context-dependent efficacy of CAF and CN supplementation. Exploring their impacts on muscle fibers, such as slow twitch vs. fast twitch, is imperative during varied resistance training exercises, including eccentric, concentric, and isometric contractions. In their systematic review, Marinho et al. (2023) highlighted that the combination of CAF and CN could provide additional benefits to exercise performance, especially when CAF is ingested after a CN loading period, indicating the need to explore these effects in various exercise modalities [4].

Longitudinal studies would be valuable for understanding how these supplements influence muscle growth, intramuscular signaling pathways, and hormonal responses. The study by Tarnopolsky (2011) on CAF and creatine use in sports underscores their ergogenic potential in specific sports contexts, suggesting that their long-term effects warrant further investigation [57]. The study of neuromuscular efficiency and the rate of force development in relation to these supplements is crucial. This is important for understanding muscle fiber recruitment patterns across different athletic populations and training statuses.

### 4.2. Secondary Outcomes: Cardiovascular Function and Blood Chemistry

Regarding the safety profile of CN and CAF, our study focused on their potential cardiovascular function and blood chemistry impacts in resistance-trained athletes. The absence of significant changes in HR and BP pre- and post-exercise supports the short-term cardiovascular safety of these supplements. This aligns with a study on rats’ lean body mass composition, which examined high doses of creatine and CAF supplementation, supporting the safety of these supplements [58].

The crossover design of our study, where we supplemented the same athletes with different substances week after week, was strategic to isolate and compare the effects of each supplementation regimen directly. This approach minimized inter-individual variability and allowed a more subtle understanding of how each substance uniquely influenced performance and cognitive function. Regarding the washout period, we employed seven days, shorter than the ideal four weeks [59], due to practical constraints to balance thorough washout with study feasibility and participant retention. Moreover, our study’s limitations extend beyond the washout period, encompassing its focus on resistance-trained male athletes, which may not fully represent broader or gender-diverse populations. While sufficient for statistical analysis, the sample size might not capture individual variability. We also did not assess certain parameters like peak blood caffeine, nitrate levels, and muscle creatine content, which, along with the lack of strict dietary control and reliance on self-reporting, could affect the outcomes or mask the nuanced effects of supplementation.

## 5. Conclusions

This study offers evidence of the cognitive benefits of co-ingestion of caffeine and creatine nitrate in resistance-trained athletes. While these supplements were shown to boost cognitive function, they did not yield significant improvements in exercise performance. This study also contributes to the sports nutrition field by confirming the short-term safety of these supplements. Future research is warranted to comprehensively explore these supplements over a longer duration and across a more diverse demographic to ascertain their role in cognitive and exercise performance.

## Figures and Tables

**Figure 1 nutrients-16-00766-f001:**
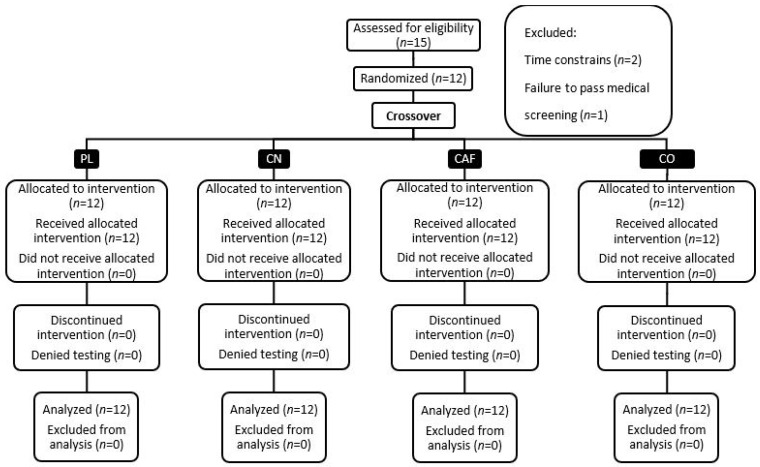
Diagram depicting the consolidated standards for reporting trials in the study.

**Figure 2 nutrients-16-00766-f002:**
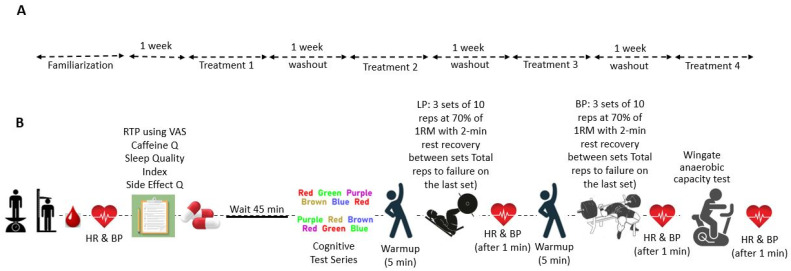
Study process overview. Panel (**A**) illustrates the study’s timeline, detailing the familiarization period followed by four treatment phases, each separated by a washout period of one week. Panel (**B**) provides a detailed sequence of assessment and intervention activities conducted during each treatment phase.

**Figure 3 nutrients-16-00766-f003:**
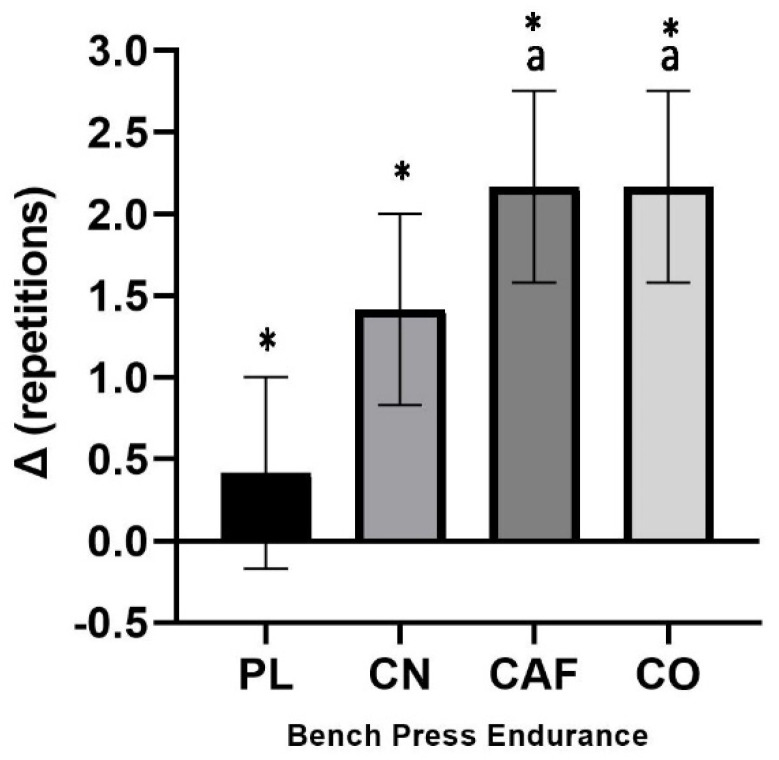
The mean change (with a 95% confidence interval) in the number of bench press repetitions to failure at 70% of 1RM from the initial measurement. Statistically significant findings (*p* < 0.05) are indicated by confidence intervals that do not cross zero. * denotes a significant difference (*p* < 0.05) compared to the baseline. We use the following notation to indicate statistical differences between treatments: (a) a significant difference compared to PL (*p* = 0.04).

**Figure 4 nutrients-16-00766-f004:**
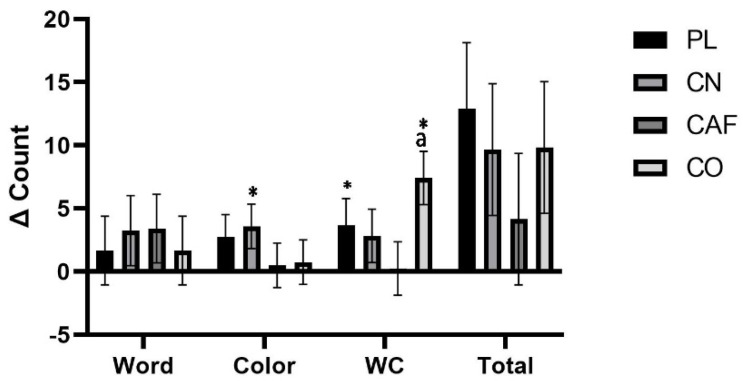
The graph illustrates the mean change (95% CI) from baseline to follow-up for the Stroop test-related treatment. Statistically significant findings (*p* < 0.05) are indicated by confidence intervals that do not cross zero. * denotes a significant difference (*p* < 0.05) compared to the baseline. Statistical significance is shown as follows: (a) significant difference compared to CAF (*p* = 0.04).

**Table 1 nutrients-16-00766-t001:** Baseline demographics.

Measurement	Mean SD
Age (year)	21.9 ± 0.79
Height (cm)	179.9 ± 9.53
Body Mass Index (kg/m^2^)	26.6 ± 5.11
Bench press 1RM (kg)	101.1 ± 27.8
Leg press 1RM (kg)	371.4 ± 64.7

Mean data are presented as means ± SD. 1RM: one repetition maximum.

**Table 2 nutrients-16-00766-t002:** Dietary analysis.

Variable	Treatment	Mean SD	*p*-Value
Energy Intake (kcal/d/kg)	PL	24.1 ± 9.15	0.07
	CN	21.9 ± 7.22	
	CAF	23.2 ± 8.27	
	CO	20.4 ± 5.12	
Protein (g/d/kg)	PL	1.34 ± 0.52	0.08
	CN	1.26 ± 0.50	
	CAF	1.32 ± 0.35	
	CO	1.10 ± 0.37	
Carbohydrate (g/d/kg)	PL	2.24 ± 1.24	0.43
	CN	2.16 ± 0.90	
	CAF	2.34 ± 1.38	
	CO	1.99 ± 0.61	
Fat (g/d/kg)	PL	1.01 ± 0.42	0.09
	CN	0.86 ± 0.41	
	CAF	0.83 ± 0.45	
	CO	0.79 ± 0.35	

Values are means ± standard deviations. Multivariate analysis revealed overall Wilks’ Lambda for treatment (*p* = 0.78).

**Table 3 nutrients-16-00766-t003:** Exercise performance associated with supplementation.

Variable	Treatment	Day 0	Day 7	Treatment		
		Mean SD	Mean SD	Mean SE	*p*-Value	
Lifting repetitions and volume						
BP Repetitions to failure at 70% 1RM	PL	10.3 ± 2.71	10.7 ± 2.34 ^c,d^	10.5 ± 0.51	Time	0.02
	CN	10.5 ± 2.93	11.9 ± 3.55 *	11.2 ± 0.66	Trt	0.01
	CAF	11.5 ± 3.84	13.7 ± 3.76 ^a,^*	12.6 ± 0.79	T×T	0.43
	CO	11.7 ± 3.64	13.9 ± 4.64 ^a,^*	12.8 ± 0.86		
BP third set lifting volume (kg)	PL	711.6± 239.1	734.5 ± 219.7	723.1 ± 45.9	Time	0.007
	CN	695.3 ± 168.2	804.4 ± 284.8 *	749.9 ± 48.1	Trt	0.02
	CAF	789.8 ± 284.2	941.9 ± 289.6 *	865.9 ± 59.4	T×T	0.50
	CO	793.4 ± 249.8	949.1 ± 332.7 *	871.3 ± 60.9		
LP Repetitions to failure at 70% 1RM	PL	28.1 ± 9.57	29.4 ± 8.96	28.7 ± 1.85	Time	0.06
	CN	28.2 ± 9.71	32.7 ± 10.8	30.5 ± 2.11	Trt	0.16
	CAF	33.4 ± 16.4	32.4 ± 16.9	32.9 ± 3.33	T×T	0.86
	CO	33.9 ± 14.8	38.5 ± 15.8	36.2 ± 3.11		
LP third set lifting volume (kg)	PL	7935.1 ± 4139.5	7715.7 ± 3394.4	7825.4 ± 756.1	Time	0.12
	CN	7328.9 ± 2923.5	8473.8 ± 3236.8	7901.4 ± 627.2	Trt	0.28
	CAF	8735.1 ± 4474.6	8435.8 ± 4675.6	8585.4 ± 914.1	T×T	0.79
	CO	8906.7 ± 4401.7	10,024.3 ± 4623.8	9465.5 ± 908.7		
Combined lifting volume (kg)	PL	14,443.4 ± 4684.9	14,966.5 ± 4463.0	14,705.0 ± 915.0	Time	0.07
	CN	14,540.5 ± 3922.7	15,794.5 ± 4247.7	15,167.5 ± 826.6	Trt	0.12
	CAF	15,669.9 ± 6258.7	15,727.7 ± 5706.3	15,698.8 ± 1195.6	T×T	0.80
	CO	16,216.4 ± 5355.6	17,489.7 ± 5489.4	16,853.1 ± 1090.7		
Anaerobic capacity test Wingate test						
Total work (J)	PL	19,394.1 ± 3032.1	19,742.7 ± 2791.9 ^c^	19,568.4 ± 582.9	Time	0.40
	CN	19,555.6 ± 2639.2	19,999.3 ± 2812.3 ^c^	19,777.5 ±546.4	Trt	0.54
	CAF	20,039.2 ± 2368.4	19,341.1 ± 2257.1 ^a,b,d^	19,690.1 ± 467.5	T×T	0.20
	CO	19,111.5 ± 2710.1	19,373.4 ± 2883.6 ^c^	19,242.5 ± 559.3		
Mean power (W)	PL	671.7 ± 99.7	677.7 ± 87.9 ^c^	674.7 ± 18.7	Time	0.31
	CN	681.2 ± 89.7	698.1 ± 98.2 ^c^	689.6 ± 18.8	Trt	0.35
	CAF	688.5 ± 82.7	663.4 ± 78.5 ^a,b,d^	675.9 ± 16.3	T×T	0.15
	CO	659.8 ± 92.3	673.3 ± 105.5 ^c^	666.5 ± 19.8		
Peak power (W)	PL	1021.3 ± 174.1	1025.0 ± 173.5	1023.2 ± 34.7	Time	0.01
	CN	1026.6 ± 186.4	1041.2 ± 180.4	1033.9 ± 36.6	Trt	0.78
	CAF	991.8 ± 189.0	956.7 ± 176.1	974.2 ± 36.6	T×T	0.67
	CO	941.3 ± 136.1	969.0 ± 180.7	962.8 ± 35.6		
Minimum power (W)	PL	389.1 ± 69.5	396.5 ± 95.3	392.8 ± 16.6	Time	0.09
	CN	356.4 ± 91.3	366.4 ± 88.6	361.4 ± 18.0	Trt	0.09
	CAF	410.2 ± 75.9	413.6 ± 70.6	411.9 ± 14.6	T×T	0.75
	CO	396.7 ± 70.5	380.1 ± 97.9	388.4 ± 17.1		
Fatigue index (%)	PL	61.6 ± 6.41	59.7 ± 12.1	60.7 ± 1.94	Time	0.01
	CN	64.9 ± 8.63	64.4 ± 8.66	64.6 ± 1.72	Trt	0.21
	CAF	57.6 ± 9.52	55.7 ± 10.9	56.6 ± 2.05	T×T	0.74
	CO	57.3 ± 9.06	59.3 ± 14.2	58.3 ± 2.39		

Values are means ± standard deviations. Multivariate analysis revealed overall Wilks’ Lambda time (*p* = 0.49), treatment (*p* = 0.04), and time × treatment (*p* = 0.99). *p*-values for the Greenhouse–Geisser time and the time-by-treatment (T×T) interaction are presented alongside univariate group *p*-values. ^a^ denotes a significant difference from PL. ^b^ denotes a significant difference from CN. ^c^ denotes a significant difference from CAF. ^d^ denotes a significant difference from CO. * represents *p* < 0.05 difference from baseline. Trt = treatment, T×T = time × treatment interaction.

**Table 4 nutrients-16-00766-t004:** Cognitive function.

Variable	Treatment	Day 0	Day 7	Treatment		
		Mean SD	Mean SD	Mean SE	*p*-Value	
Word (counts)	PL	109.3 ± 17.7	111.0 ± 13.4	110.1 ± 3.14	Time	0.006
	CN	113.2 ± 12.7	116.0 ± 14.2	114.8 ± 2.72	Trt	0.15
	CAF	116.1 ± 14.5	119.5 ± 17.3	117.7 ± 3.21	T×T	0.55
	CO	118.1 ± 20.3	119.7 ± 16.4	118.9 ± 3.69		
Color (counts)	PL	83.6 ± 16.2	86.41 ± 15.1	85.0 ± 3.14	Time	0.003
	CN	83.4 ± 13.4	87.0 ± 13.8 *	85.2 ± 2.74	Trt	0.26
	CAF	92.1 ± 15.5	92.5 ± 16.6	92.3 ± 3.21	T×T	0.64
	CO	90.6 ± 17.3	91.4 ± 17.2	91.0 ± 3.45		
Word–Color (counts)	PL	51.3 ± 18.2	59.8 ± 14.3 ^b,^*	55.5 ± 3.39	Time	0.01
	CN	54.0 ± 11.2	56.8 ± 10.8	55.4 ± 2.22	Trt	0.03
	CAF	62.5 ± 14.8	62.7 ± 14.8 ^a,c^	62.6 ± 2.95	T×T	0.04 †
	CO	58.5 ± 18.2	65.9 ± 17.3 ^b,^*	62.2 ± 3.63		
Total Stroop Results (counts)	PL	244.3 ± 48.1	257.2 ± 40.6 *	250.7 ± 8.98	Time	0.003
	CN	250.6 ± 35.5	260.3 ± 35.1	255.5 ± 7.12	Trt	0.03
	CAF	270.6 ± 38.7	274.8 ± 46.2	272.7 ± 8.52	T×T	0.67
	CO	267.2 ± 53.8	277.1 ± 46.9	273.1 ± 10.2		

Values are means ± standard deviations. Multivariate analysis revealed overall Wilks’ Lambda time (*p* = 0.71), treatment (*p* = 0.34), and time × treatment (*p* = 0.97). *p*-values for the Greenhouse–Geisser time and the time-by-treatment (T×T) interaction are presented alongside univariate group *p*-values. † indicates a significant between-treatment difference, * represents *p* < 0.05 difference from baseline. ^a^ denotes a significant difference from PL. ^b^ denotes a significant difference from CAF. ^c^ denotes a significant difference from CO. Trt = treatment, T×T = time × treatment interaction.

**Table 5 nutrients-16-00766-t005:** Muscle and liver enzymes and markers of catabolism.

Variable	Treatment	Day 0	Day 7		
		Mean SD	Mean SD	*p*-Value	
ALP (U/L)	PL	69.1 ± 25.6	69.0 ± 22.2	Time	0.63
	CN	67.8 ± 22.6	70.7 ± 25.8	Trt	0.32
	CAF	69.4 ± 24.2	67.7 ± 24.4	T×T	0.86
	CO	68.1 ± 23.2	69.5 ± 25.7		
ALT (U/L)	PL	21.5 ± 9.98	22.5 ± 11.7	Time	0.48
	CN	21.9 ± 14.3	21.0 ± 11.1	Trt	0.62
	CAF	23.4 ± 16.0	20.9 ± 10.2	T×T	0.51
	CO	24.2 ± 11.2	22.4 ± 9.27		
AST (U/L)	PL	24.3 ± 7.41	25.7 ± 6.56	Time	0.48
	CN	24.1 ± 6.88	23.1 ± 5.49	Trt	0.63
	CAF	24.8 ± 7.04	27.0 ± 16.7	T×T	0.58
	CO	25.4 ± 9.69	27.8 ± 14.1		
CK (U/L)	PL	294.7 ± 203.4	300.5 ± 162.9	Time	0.67
	CN	280.1 ± 192.5	273.3 ± 265.2	Trt	0.94
	CAF	270.1 ± 174.3	273.3 ± 151.6	T×T	0.89
	CO	243.3 ± 224.3	301.8 ± 226.4		
LDH (U/L)	PL	165.1 ± 34.2	164.5 ± 29.1	Time	0.70
	CN	168.7 ± 27.9	170.3 ± 30.4	Trt	0.39
	CAF	160.5 ± 26.5	169.9 ± 32.6	T×T	0.33
	CO	161.5 ± 26.4	169.2 ± 26.7		
BUN (mg/dL)	PL	17.5 ± 3.59	17.9 ± 4.82	Time	0.67
	CN	17.1 ± 5.07	18.1 ± 2.57	Trt	0.69
	CAF	18.8 ± 3.05	17.3 ± 3.08	T×T	0.37
	CO	16.2 ± 3.70	17.2 ± 3.86		
Creatinine (mg/dL)	PL	1.39 ± 0.24	1.42 ± 0.30	Time	0.13
	CN	1.32 ± 0.24	1.33 ± 0.25	Trt	0.62
	CAF	1.23 ± 0.21	1.30 ± 0.17	T×T	0.29
	CO	1.43 ± 0.31	1.25 ± 0.13		
BUN/Creatinine	PL	13.1 ± 4.44	13.1 ± 4.42	Time	0.56
	CN	13.4 ± 4.67	14.1 ± 3.11	Trt	0.51
	CAF	15.7 ± 3.79	13.3 ± 2.33	T×T	0.14
	CO	11.8 ± 3.81	13.9 ± 3.68		

Values are means ± standard deviations for alkaline phosphatase (ALP), aspartate transaminase (AST), alanine transaminase (ALT), creatine kinase (CK), lactate dehydrogenase (LDH), blood urea nitrogen (BUN), and creatinine. MANOVA analysis revealed overall Wilks’ Lambda time (*p* = 0.68), treatment (*p* = 0.99), and time × treatment (*p* = 0.98) effects. *p*-values for the Greenhouse–Geisser time and the time-by-treatment (T×T) interaction are presented alongside univariate group *p*-values.

**Table 6 nutrients-16-00766-t006:** Blood lipid panel.

Variable	Treatment	Day 0	Day 7		
		Mean SD	Mean SD	*p*-Value	
Cholesterol (mg/dL)	PL	172.5 ± 51.1	163.0 ± 38.5	Time	0.72
	CN	172.8 ± 52.4	172.1 ± 46.2	Trt	0.19
	CAF	167.5 ± 45.1	165.6 ± 42.5	T×T	0.56
	CO	173.6 ± 41.7	169.9 ± 48.2		
HDL-c (mg/dL)	PL	52.8 ± 12.7	51.0 ± 11.6 ^c^	Time	0.27
	CN	52.2 ± 10.3	52.8 ± 10.4	Trt	0.10
	CAF	50.9 ± 11.7	49.8 ± 13.1 ^c^	T×T	0.36
	CO	50.6 ± 12.4	53.8 ± 11.6 *^,a^		
TC: HDL-c	PL	3.43 ± 1.26	3.35 ± 1.08	Time	0.14
	CN	3.44 ± 1.22	3.37 ± 1.05	Trt	0.19
	CAF	3.51 ± 1.48	3.52 ± 1.25 ^c^	T×T	0.13
	CO	3.69 ± 1.48	3.37 ± 1.11 ^b^		
LDL-c (mg/dL)	PL	104.6 ± 48.2	97.7 ± 37.3	Time	0.43
	CN	107.4 ± 50.4	104.9 ± 44.5	Trt	0.06
	CAF	97.5 ± 42.1	101.3 ± 40.8	T×T	0.21
	CO	107.0 ± 40.1	101.2 ± 44.9		
TG (mg/dL)	PL	78.7 ± 29.1	82.1 ± 34.3	Time	0.43
	CN	81.1 ± 32.3	76.2 ± 26.3	Trt	0.29
	CAF	102.1 ± 76.7	75.8 ± 40.6	T×T	0.31
	CO	83.1 ± 28.3	77.0 ± 36.4		

Values are means ± standard deviations for total cholesterol (TC), high-density lipoproteins (HDL-c), the CHOL: HDL ratio (TC: HDL-c), low-density lipoproteins (LDL-c), and triglycerides (TG). MANOVA analysis revealed overall Wilks’ Lambda treatment (*p* = 0.97), time (*p* = 0.92), and treatment × time (*p* = 0.91). *p*-values for the Greenhouse–Geisser time and the time-by-treatment (T×T) interaction are presented alongside univariate group *p*-values. * represents *p* < 0.05 difference from baseline. ^a^ denotes a significant difference from PL. ^b^ denotes a significant difference from CAF. ^c^ denotes a significant difference from CO. Trt = treatment, T×T = time × treatment interaction.

**Table 7 nutrients-16-00766-t007:** Complete blood count data.

Variable	Treatment	Day 0	Day 7		
		Mean SD	Mean SD	*p*-Value	
WBC Count (×10^3^/μL)	PL	6.20 ± 1.71	6.15 ± 1.51	Time	0.35
	CN	5.96 ± 1.39	6.46 ± 1.83	Trt	0.30
	CAF	6.29 ± 1.45	6.11 ± 1.74	T×T	0.89
	CO	6.45 ± 1.45	6.33 ± 1.61		
RBC Count (×10^6^/μL)	PL	4.99 ± 0.92	5.19 ± 0.21	Time	0.92
	CN	5.25 ± 0.26	5.22 ± 0.21	Trt	0.45
	CAF	5.19 ± 0.27	5.14 ± 0.18	T×T	0.41
	CO	5.22 ± 0.23	5.13 ± 0.24		
Hemoglobin (g/dL)	PL	16.2 ± 1.73	15.7 ± 0.41	Time	0.09
	CN	15.9 ± 0.74	15.9 ± 0.71	Trt	0.33
	CAF	15.9 ± 0.67	15.7 ± 0.73	T×T	0.78
	CO	15.7 ± 0.65	15.6 ± 0.61		
Hematocrit (%)	PL	49.1 ± 4.72	47.9 ± 1.61	Time	0.04
	CN	48.3 ± 1.91	48.4 ± 2.11	Trt	0.42
	CAF	48.1 ± 1.79	47.4 ± 1.96	T×T	0.92
	CO	48.0 ± 1.61	47.5 ± 1.84		
MCV (fL)	PL	92.7 ± 3.98	92.6 ± 3.85 ^b^	Time	0.34
	CN	92.3 ± 3.91	92.7 ± 3.80 ^a,c^	Trt	0.07
	CAF	92.7 ± 3.87	92.3 ± 3.59 ^b,d^	T×T	0.64
	CO	92.2 ± 3.68	92.6 ± 3.55 ^c^		
MCH (pg/cell)	PL	30.7 ± 1.61	30.3 ± 1.53 ^b,d^	Time	0.61
	CN	30.4 ± 1.37	30.5 ± 1.41 ^a^	Trt	0.01
	CAF	30.7 ± 1.57	30.6 ± 1.48	T×T	0.60
	CO	30.3 ± 1.51	30.6 ± 1.42 ^a^		
MCHC (g/dL)	PL	33.1 ± 0.57	32.7 ± 0.61	Time	0.28
	CN	32.9 ± 0.48	32.9 ± 0.50	Trt	0.34
	CAF	33.1 ± 0.71	33.1 ± 0.62	T×T	0.39
	CO	32.9 ± 0.58	33.0 ± 0.51		
RDW (%)	PL	12.8 ± 0.69	12.6 ± 0.66	Time	0.77
	CN	12.7 ± 0.47	12.6 ± 0.68	Trt	0.64
	CAF	12.7 ± 0.81	12.6 ± 0.95	T×T	0.91
	CO	12.7 ± 0.51	12.7 ± 0.64		
Platelet Count (×10^3^/μL)	PL	241.7 ± 55.5	234.1 ± 46.5	Time	0.33
	CN	262.3 ± 43.3	257.6 ± 42.8	Trt	0.002
	CAF	252.5 ± 45.5	244.1 ± 41.4	T×T	0.47
	CO	261.9 ± 38.6	254.1 ± 40.9		

Values are means ± standard deviations. All variables were analyzed by MANOVA. MANOVA analysis revealed overall Wilks’ Lambda time (*p* = 0.43), treatment (*p* = 0.92), and time × treatment (*p* = 0.97). *p*-values for the Greenhouse–Geisser time and the time-by-treatment (T×T) interaction are presented alongside univariate group *p*-values. ^a^ denotes a significant difference from PL. ^b^ denotes a significant difference from CN. ^c^ denotes a significant difference from CAF. ^d^ denotes a significant difference from CO. Trt = treatment, T×T = time × treatment interaction.

## Data Availability

Data are contained within the article.

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
