# Peer review of "The Effect of Creatine Nitrate and Caffeine Individually or Combined on Exercise Performance and Cognitive Function: A Randomized, Crossover, Double-Blind, Placebo-Controlled Trial"

_nutrients, 2024, doi:10.3390/nu16060766_

Round 1
Reviewer 1 Report
Comments and Suggestions for Authors
Dear authors, please find my minor comments below.
Lines 90–93: provide a description of the primary aim and secondary outcomes, then organise results and discussion in this sense.
Lines 143–145: Regarding pharmacokinetics, peak caffeine plasma concentrations after oral administration are reported to occur at times ranging from 30 to 120 min, but the most common timing is 60 min before exercise (doi.org/10.1186/s12970-020-00383-4). In addition, peak plasma creatine concentration is achieved 1 hour after ingestion (doi:10.1186/1550-2783-4-17). In this sense, it should be better to wait more to optimise timing ingestion before performance (see here: doi.org/10.3389/fnut.2022.887523).
Lines 183–184: “Body fat percentage was ascertained using the HBF-306C handheld bioelectric impedance analysis device from Omron Healthcare." This is not relevant in your manuscript; delete it or add to the limitation that you are not using a gold standard (as dexa) to assess body composition. In addition, the predictive equation to estimate body fat is not provided, and the use of specific predictive equations results in better FM% estimation (doi.org/10.3390/nu15020278).
Author Response
Lines 90–93: provide a description of the primary aim and secondary outcomes, then organise results and discussion in this sense.
Based on your request, we have organized the manuscript to align with the reviewer's instructions, focusing on the primary aim and secondary outcomes as outlined. We have structured the content to address the investigation of the ergogenic value of caffeine, creatine nitrate, and their co-ingestion over seven days on intense intermittent exercise performance, mental focus, cardiovascular hemodynamics, and blood metabolites in resistance-trained males.
Lines 143–145: Regarding pharmacokinetics, peak caffeine plasma concentrations after oral administration are reported to occur at times ranging from 30 to 120 min, but the most common timing is 60 min before exercise (doi.org/10.1186/s12970-020-00383-4). In addition, peak plasma creatine concentration is achieved 1 hour after ingestion (doi:10.1186/1550-2783-4-17). In this sense, it should be better to wait more to optimise timing ingestion before performance (see here: doi.org/10.3389/fnut.2022.887523).
We acknowledge the limitations associated with the chosen timing of supplement administration. While the 45-minute wait aimed to balance practical considerations with potential absorption, it might not have captured the peak effects of the supplements on exercise performance. Future studies could explore different time points to optimize the timing of supplement intake and exercise protocol for this population.
Lines 183–184: “Body fat percentage was ascertained using the HBF-306C handheld bioelectric impedance analysis device from Omron Healthcare." This is not relevant in your manuscript; delete it or add to the limitation that you are not using a gold standard (as dexa) to assess body composition. In addition, the predictive equation to estimate body fat is not provided, and the use of specific predictive equations results in better FM% estimation (doi.org/10.3390/nu15020278).
Thank you for your feedback. We have removed the mention of the HBF-306C handheld bioelectric impedance analysis device from Omron Healthcare from our manuscript, as suggested.
Reviewer 2 Report
Comments and Suggestions for Authors
1. The title can be changed to: The effect of creatine nitrate......; the same in the abstract aims
2. Figure 2 showing the study design is not clear for me - i think that it should be devided into A and B panels, (and please describe both panels in figure legend), because (if I understand it well) the upper diagram is not related to the bottom panel in case of time of the study stages of experiment. For me this figure was quite confusing.
3. Discussion. Please, discuss the model - if it would be better to include 4 groups of athletes or to intruduce different supplementation week after week in the same group of athletes? Provide the washout time for all of the supplements based on the literature. Do You think that this experimental model can interfere with the results of other type of supplementation introduced earlier?
4. Lines 52-53 - for clarity, phosphocreatine plays a role in rapidly replenishing adenosine tri-phosphate (ATP)
5. Line 91 - please add (1)
6. In line 479 I can read that "our study did not observe an enhancement in performance" (after supplementation). But the figure 4 shows the opposite?
Author Response
- The title can be changed to: The effect of creatine nitrate......; the same in the abstract aims
Answer: Thank you for your valuable suggestion. We have updated both the title and the abstract to incorporate the recommended changes.
- Figure 2 showing the study design is not clear for me - i think that it should be devided into A and B panels, (and please describe both panels in figure legend), because (if I understand it well) the upper diagram is not related to the bottom panel in case of time of the study stages of experiment. For me this figure was quite confusing.
Answer: Thank you for the feedback on Figure 2. We acknowledge the confusion and separated the figure into panels A and B for clarity, with a detailed description for each panel in the figure legend to depict the distinct stages and timelines of the study design.
- Discussion. Please, discuss the model - if it would be better to include 4 groups of athletes or to intruduce different supplementation week after week in the same group of athletes?
Answer: The crossover design used in this study allowed for a more controlled and efficient comparison of the effects of different supplementation regimens on the same group of athletes. Introducing different supplements week after week within the same group minimized the variability that could arise from differences in individual responses, training backgrounds, and physiological characteristics. This approach also reduced the number of participants needed to achieve statistically significant results, thus enhancing the study's feasibility and ethical alignment by limiting the subject pool. In addition, this design enabled a direct and individualized assessment of how each supplement affected performance and cognitive function over time, providing clearer insights into each supplementation strategy's specific benefits or drawbacks. This methodological choice ensured a more robust and precise understanding of the supplements' effects, which might not have been as discernible with a larger, more heterogeneous sample spread across multiple groups.
Provide the washout time for all of the supplements based on the literature.
Answer:
- The washout period for creatine nitrate is established at seven days, based on findings from sources [1, 2]. This duration is meticulously chosen to eliminate residual effects before transitioning to subsequent supplementation phases.
- Similarly, caffeine is assigned a seven-day washout period, as corroborated by studies referenced in [3, 4]. This interval is designed to reset the physiological state of participants, allowing for an unbiased assessment of subsequent interventions.
- The combined ingestion of creatine nitrate and caffeine necessitates a seven-day washout [5]. This precautionary measure ensures the clear delineation of the effects of each supplementation phase, allowing for accurate comparisons across the study's duration.
Do You think that this experimental model can interfere with the results of other type of supplementation introduced earlier?
Answer: This model does not interfere with the results of the other type of supplementation introduced earlier because it is counterbalanced. Thus, the data is collected for the first supplement type after the first administration. If there were lasting effects beyond the washout period, they have not been observed or reported to this point [6].
- Lines 52-53 - for clarity, phosphocreatine plays a role in rapidly replenishing adenosine tri-phosphate (ATP)
Answer: We have revised lines 52-53 to include the specified clarification. Thank you for the suggestion.
- Line 91 - please add (1)
Answer: We have incorporated the suggested addition at line 91. Thank you for the guidance.
- In line 479 I can read that "our study did not observe an enhancement in performance" (after supplementation). But the figure 4 shows the opposite?
Answer: Thank you for highlighting this potential contradiction. We understand the confusion. While the statement in the discussion section aimed to reflect the nuances observed in the findings, we acknowledge that it failed to do so and could easily be misconstrued.
Our interpretation is that the study showed mixed results. While significant improvements were observed for some measures (e.g., BP reps in CAF/CO, CAF's total work/mean power), there was no consistent enhancement across all groups and outcomes. For example, no significant differences from PL were found for LP reps or total work. We have amended the text to reflect this nuance and believe that it better reflects the Figure and is more cohesive with the subsequent description in the context of previous findings.
References
- Galvan, E., et al., Acute and chronic safety and efficacy of dose dependent creatine nitrate supplementation and exercise performance. J Int Soc Sports Nutr, 2016. 13: p. 12.
- Ostojic, S.M., et al., Searching for a better formulation to enhance muscle bioenergetics: A randomized controlled trial of creatine nitrate plus creatinine vs. creatine nitrate vs. creatine monohydrate in healthy men. Food Science & Nutrition, 2019. 7(11): p. 3766-3773.
- Desbrow, B., et al., The effects of different doses of caffeine on endurance cycling time trial performance. Journal of sports sciences, 2012. 30(2): p. 115-120.
- Härtter, S., et al., Effect of caffeine intake 12 or 24 hours prior to melatonin intake and CYP1A2* 1F polymorphism on CYP1A2 phenotyping by melatonin. Basic & clinical pharmacology & toxicology, 2006. 99(4): p. 300-304.
- Prins, P.J., et al., The Effect of Caffeine Alone or as Part of a Multi-ingredient Pre-workout Supplement on Muscular Endurance in Recreationally Active College Males. Journal of Exercise and Nutrition, 2018. 1(5).
- Kreider, R.B., R. Jäger, and M. Purpura, Bioavailability, efficacy, safety, and regulatory status of creatine and related compounds: A critical review. Nutrients, 2022. 14(5): p. 1035.
Round 2
Reviewer 1 Report
Comments and Suggestions for Authors
well done
Author Response
Thank you very much for your positive feedback. We are pleased to hear that the revisions met your expectations and appreciate your acknowledgment.
Reviewer 2 Report
Comments and Suggestions for Authors
Thank you for improving your manuscript according to my comments. However, I think that the explanations why you decided to supplement the same athletes by different type of substances week after week and about the washout time of supplements should be added to the discussion in the manuscript.
Author Response
We are grateful for your insightful feedback. Based on your suggestions, we have refined the relevant section of the "Discussion" as follows:
"The crossover design of our study, where we supplemented the same athletes with different substances week after week, was strategic to isolate and compare the effects of each supplementation regimen directly. This approach minimized inter-individual variability and allowed a more subtle understanding of how each substance uniquely influenced performance and cognitive function. Regarding the washout period, we employed seven days, shorter than the ideal four weeks [59], due to practical constraints to balance thorough washout with study feasibility and participant retention. Moreover, our study's limitations extend beyond the washout period, encompassing its focus on resistance-trained male athletes, which may not fully represent broader or gender-diverse populations. While sufficient for statistical analysis, the sample size might not capture individual variability. We also did not assess certain parameters like peak blood caffeine, nitrate levels, and muscle creatine content, which, along with the lack of strict dietary control and reliance on self-reporting, could affect the outcomes or mask the nuanced effects of supplementation."
We trust this amendment adequately addresses your concerns and enhances the clarity and depth of our discussion.